astrophysics/atmospheric science/plasma physics

ionosphere, sporadic E, global positioning system, global navigation satellite system, radio occultations, FORMOSAT-3/COSMIC

**Author for correspondence:**
Bingkun Yu
e-mail: bingkun.yu@reading.ac.uk

# Derivation of global ionospheric Sporadic E critical frequency ($f_o$Es) data from the amplitude variations in GPS/GNSS radio occultations

Bingkun Yu[1,2], Christopher J. Scott[1], Xianghui Xue[2,3,4,5], Xinan Yue[6] and Xiankang Dou[2,4,7]

[1]Department of Meteorology, University of Reading, Reading RG6 6BB, UK
[2]CAS Key Laboratory of Geospace Environment, Department of Geophysics and Planetary Sciences, [3]Anhui Mengcheng Geophysics National Observation and Research Station and [4]Hefei National Laboratory for the Physical Sciences at the Microscale, University of Science and Technology of China, Hefei 230026, People's Republic of China
[5]CAS Center for Excellence in Comparative Planetology, Hefei 230026, People's Republic of China
[6]Key Laboratory of Earth and Planetary Physics, Institute of Geology and Geophysics, Chinese Academy of Sciences, Beijing 100029, People's Republic of China
[7]Wuhan University, Wuhan 430072, People's Republic of China

BY, 0000-0003-2758-1960; CJS, 0000-0001-6411-5649; XX, 0000-0002-4541-9900; XY, 0000-0003-3379-9392; XD, 0000-0001-6433-6222

The ionospheric sporadic E (Es) layer has a significant impact on the global positioning system (GPS)/global navigation satellite system (GNSS) signals. These influences on the GPS/GNSS signals can also be used to study the occurrence and characteristics of the Es layer on a global scale. In this paper, 5.8 million radio occultation (RO) profiles from the FORMOSAT-3/COSMIC satellite mission and ground-based observations of Es layers recorded by 25 ionospheric monitoring stations and held at the UK Solar System Data Centre at the Rutherford Appleton Laboratory and the Chinese Meridian Project were used to derive the hourly Es critical frequency ($f_o$Es) data. The global distribution of $f_o$Es with a high spatial resolution shows a strong seasonal variation in $f_o$Es with a summer maximum exceeding 4.0 MHz and a winter minimum between 2.0 and 2.5 MHz. The GPS/GNSS RO technique is an important tool that can provide global estimates of Es layers, augmenting the limited coverage and low-frequency detection threshold of ground-based instruments. Attention should be paid to small $f_o$Es values from ionosondes near the instrumental detection limits corresponding to minimum frequencies in the range 1.28–1.60 MHz.

# 1. Introduction

The term 'sporadic E (Es)' was first used to describe the abnormal nocturnal E layer at nearly 105 km in the early 1930s when the term 'ionosphere' was introduced by Watson–Watt at the UK's Radio Research Station to designate the ionized layers in the Earth's upper atmosphere [1,2]. These previous records suggest that the occurrence of Es layers (nocturnal E-region ionization) is correlated with not only the occurrence of magnetic storms but also the occurrence of thunderstorms. An enhancement of the ionospheric Es layer due to lightning was later reported by a statistical superposed epoch analysis (SEA) [3] and confirmed by following studies [4–11]. The time delay between thunderstorm activities and the response of Es layers is associated with the tidal periodicities in the Es variability [12,13]. For the mid-latitude, the most widely accepted mechanism for the production of Es layers is the wind shear theory [14,15]. Patches of ionization within Es layers form as a result of the vertical ion convergence driven by vertical shears in the zonal neutral wind (a westward wind increasing with altitude) and meridional neutral wind (a northward wind increasing with altitude for the Northern Hemisphere, or a southward wind increasing with altitude in the Southern Hemisphere). The wind shear theory indicates that the formation of Es layers should be inhibited on the magnetic equator, because the ions fail to converge vertically into a layer when the magnetic field is horizontal [16]. The equatorial Es arises from the gradient instability and is associated with the enhanced electro-jet current [17,18]. Based on the expression for the vertical ion drift in wind shear theory [16], the vertical velocity of ions is very small when $cosI \sim 0$. Here, $I$ denotes the magnetic dip angle. So the ion-convergence mechanism does not work efficiently at high magnetic latitudes [19], but the vertical effects of gravity waves and electric field are very efficient in concentrating the ionization of Es layers in the central polar cap where the magnetic-field lines are approximately vertical [20,21]. The wind shear theory has been confirmed by incoherent scatter radar (ISR) [22], meteor radar [23], ionosonde [24], rocket [25], lidar [26] and chemical tracer measurements [27], as well as the measurements [28–34] and model simulations [35–40] of metal ions and atoms in the mesosphere and lower thermosphere (MLT). The main problem of wind shear theory is to explain the global morphology of Es and its seasonal distribution [16,17].

Since their inception in the early 1930s, ground-based instrumentations for the radio sounding of the ionosphere have been developed and are now made routinely throughout the world [19]. The first global map of the occurrence of Es layers was produced by employing ionosonde data [41]. However, the limited number of ground-based instruments with sparse coverage made it difficult to advance the scientific understanding of the global Es layer and its formation mechanism with a high spatial resolution. Since the intense plasma irregularities within Es layers have very sharp vertical gradients in electron number density, the Es layers have serious effects on radio communications and navigation systems [42]. The Es layer contributes more than one-third of ionospheric irregularities leading to the occurrence of an interruption in the global positioning system (GPS)/global navigation satellite system (GNSS) signal tracking. These influences are crucial for the precision, accuracy, reliability and application of the modern real-time GNSS high-precision positioning [43]. The ionospheric effects on the signals of GNSS radio occultation (RO) receivers can be exploited for extracting information on the ionospheric structures of electron density irregularities [42,44]. The global occurrence of Es layers has been widely studied using GNSS RO signals from FORMOSAT-3 (FORMosa SATellite Mission-3)/COSMIC (Constellation Observing System for Meteorology, Ionosphere and Climate), GRACE (Gravity Recovery and Climate Experiment) and CHAMP (CHAllenging Minisatellite Payload) [45–51]. Model simulations show that the seasonal variation in the occurrence rate of the Es layer is potentially attributed to the convergence of the metal ions driven by wind shears [52]. A global map of the amplitude scintillation index (S4), a proxy of the intensity of Es layers, also presents a strong seasonal dependence [16]. However, one weak point in explaining the seasonal dependence of Es remains; that the geographical distribution of the Es minimum in the winter hemisphere cannot be simulated by the neutral wind shears. More extensive observations of Es layers from both ground-based instruments and satellites will help to comprehensively understand the seasonal variation in the Es layer and its mechanism. Recent studies have revealed the relationship between the S4max (maximum values of S4) and the intensity of Es layers [53,54]. The blanketing frequency of Es layers, $f_bEs$, is related to the S4max index obtained by FORMOSAT-3/COSMIC RO measurements, based on a small number of local observations.

Ionosondes provide reliable ground-based observations of the local intensity of Es layers. The highest shortwave radio frequency returned vertically from the ionospheric Es layer is referred to as the critical frequency of the layer, $f_oEs$ (in Hz). This represents the plasma frequency of the layer peak, which in turn is associated with the peak electron concentration of the Es layer, $N_e$ (in $m^{-3}$), by the formula

$f_o Es = 8.98\sqrt{N_e}$ [55]. The S4 index derived from satellite measurements, is defined as the standard deviation of signal intensity fluctuations normalized by average intensities. The S4max index is the maximum value of the amplitude scintillation S4 index in the GPS/GNSS RO signals. Large S4max values are associated with strong vertical gradients in ionospheric electron number density [56]. In this paper, global hourly ionospheric $f_o Es$ values were derived, through comparison of global GNSS-RO satellite measurements and localized ground-based ionosonde observations. Hourly coincident events were analysed using FORMOSAT-3/COSMIC RO data and ionosonde data from 25 low- to middle-latitude stations in the period 2006–2014. The relation between the two types of measurement was used to derive a high-resolution global map of the intensity of Es layers, in which the effects of the Earth's magnetic field, the diurnal and semi-diurnal tides on the latitude/longitude distribution of $f_o Es$ in both hemispheres are apparent. In addition, the quality of ground-based observations from the worldwide ionosonde network can be evaluated by comparing with the global $f_o Es$ data derived from FORMOSAT-3/COSMIC RO data.

## 2. Database

The FORMOSAT-3/COSMIC mission is a low-Earth-orbit (LEO) constellation of six microsatellites launched from Vandenberg Air Force Base in April 2006 [57]. Six FORMOSAT-3/COSMIC satellites initially followed the same orbit at approximately 512 km and then subsequently orbited the Earth at 800 km. Each satellite has four antennas: two occultation antennas for 50 Hz rate tracking to retrieve the lower atmosphere parameters (e.g. the temperature, bending angle and refractivity), and two precise orbit determination (POD) antennas for 1 Hz tracking to determine the LEO orbit and retrieve the ionospheric electron number density, slant total electron content and scintillation index [58]. The POD antennas sampled the amplitude of any scintillation at a rate of 50 Hz in the L1 band. A 1 Hz standard deviation was calculated from this onboard the spacecraft and transferred to the ground. Long-term detrended FORMOSAT-3/COSMIC S4 data were processed and archived from the signal-to-noise ratio (SNR) intensity fluctuations of the GPS/GNSS RO signals by the FORMOSAT-3/COSMIC Data Analysis and Archive Center (CDAAC) [59].

In the present study, the computed detrended S4max data occurring between 90 and 130 km altitude over a 9-year period from 2006 to 2014 were used to study the intensity of Es layers. The FORMOSAT-3/COSMIC can provide 1500–2500 RO measurements per day and a total of approximately 5.8 million S4max profiles were used to study the occurrence and intensity of Es layers during the study period.

The coincident data records of $f_o Es$ from 25 ground-based stations were taken from the UK Solar System Data Centre at the Rutherford Appleton Laboratory (UKSSDC; http://www.ukssdc.ac.uk) [60] and the Chinese Meridian Project (data.meridianproject.ac.cn) [61]. Most ground-truth observations are manually scaled or automatically scaled except the SanVito ionosonde. Table 1 lists the ionosonde stations used in analysis in order of decreasing north latitude.

For the comparison, S4max observations made within a region of 5° × 5° geographical latitudes and longitudes square centred on each ionosonde station were used. The S4max values were hourly averaged before comparison with hourly ionosonde data. A total of 26 863 h coincident events were analysed using the FORMOSAT-3/COSMIC S4max data and the ionosonde data from 25 stations.

## 3. Data analysis

Figure 1 displays the scatter plot of the relationship between the hourly intensity of Es layers characterized by S4max from FORMOSAT-3/COSMIC RO measurements and coincident hourly $f_o Es$ ground-based measurements for 25 ionosonde stations in the period 2006–2014. A general correlation is found between $f_o Es$ measured by the ionosondes and FORMOSAT-3/COSMIC S4max at different latitudes. It also shows that the performance of the manual-scaling algorithms for ionosonde data has a problem characterizing the intensity of Es layers when the $f_o Es$ is close to the low-frequency scaling threshold below 2.0 MHz. A difference should be noted though: the limitation of ionosonde sensitivity and the lower threshold of reliable $f_o Es$ for manual or automatic scaling. The former is the lowest frequency of reflection wave recorded in the ionogram (1.0–1.5 MHz) [19], which is dependent on the sensitivity level of the recording system and absorption in the ionosphere. The sensitivities of different ionosondes differ. The latter is the scaling threshold of reliable values in $f_o Es$ data identified as the highest shortwave radio frequency in the Es layer traces from ionograms, below which it becomes

**Table 1.** Ionosonde stations used in the analysis.

| no. | st. code | st. name | Lat. | Lon. | Mag. Lat | Mag. Lon | years | type |
|---|---|---|---|---|---|---|---|---|
| 1 | SO166 | Sodankyla | 67.40 | 26.60 | 63.90 | 119.74 | 2006–2014 | manual |
| 2 | MH453 | Mohe | 52.00 | 122.50 | 42.10 | −167.78 | 2010–2014 | manual DPS4D |
| 3 | RL052 | Chilton | 51.60 | −1.30 | 53.63 | 83.67 | 2006–2009 | automatic edited DPS-1 |
| 4 | ML449 | Manzhouli | 49.60 | 117.50 | 39.55 | −171.82 | 2008–2014 | manual |
| 5 | VT139 | SanVito | 40.70 | 17.90 | 39.75 | 98.53 | 2009 | automatic DISS |
| 6 | BP440 | Beijing | 40.30 | 116.20 | 30.22 | −172.56 | 2006–2014 | manual DPS4D |
| 7 | WU430 | Wuhan | 30.50 | 114.40 | 20.41 | −173.91 | 2010–2014 | manual DPS4D |
| 8 | EG931 | EglinAFB | 30.40 | −86.80 | 39.86 | −16.47 | 2006–2007 | manual DISS |
| 9 | 09429 | Chongqing | 29.50 | 106.40 | 19.36 | 178.72 | 2008–2014 | manual |
| 10 | SH427 | Shaoyang | 27.10 | 111.30 | 16.98 | −176.73 | 2012–2014 | manual DPS4D |
| 11 | GU421 | Guangzhou | 23.10 | 113.40 | 13.02 | −174.70 | 2008–2014 | manual |
| 12 | SA418 | Sanya | 18.30 | 109.40 | 8.21 | −178.45 | 2007–2014 | manual DPS4D |
| 13 | VA50L | Vanimo | −2.70 | 141.30 | −10.98 | −145.82 | 2006–2009 | manual |
| 14 | PY50R | PortMoresby | −9.41 | 147.15 | −16.99 | −139.13 | 2006–2007 | manual |
| 15 | CS31K | Cocosls | −12.20 | 96.80 | −21.91 | 168.41 | 2008–2014 | manual |
| 16 | DW41K | Darwin | −12.45 | 130.95 | −21.51 | −155.61 | 2006–2014 | manual DPS-4 |
| 17 | BR52P | Brisbane | −27.53 | 152.92 | −34.16 | −130.49 | 2006–2014 | manual |
| 18 | MU43K | Mundaring | −31.98 | 116.22 | −41.67 | −170.40 | 2006–2007 | manual |
| 19 | CB53N | Canberra | −35.32 | 149.00 | −42.34 | −133.21 | 2006–2014 | manual IPS-5A |
| 20 | HO54K | Hobart | −42.92 | 147.32 | −50.04 | −133.28 | 2006–2014 | manual |
| 21 | GH64L | Christchurch | −43.42 | 172.34 | −46.79 | −106.08 | 2006–2011 | manual |
| 22 | PSJ5J | Stanley | −51.70 | −57.80 | −41.79 | 11.99 | 2006–2009 | automatic edited DPS-1 |
| 23 | MQ55M | Macquarielsland | −54.50 | 159.00 | −59.69 | −115.72 | 2006–2013 | manual |
| 24 | MW26P | Mawson | −67.60 | 62.90 | −73.08 | 111.63 | 2006–2014 | manual |
| 25 | SB67Q | ScottBase | −77.90 | 166.80 | −78.97 | −70.94 | 2006–2007 | manual |

more challenging to distinguish between the Es layer and the background E layer. As an example, measurements from the ionosonde MU43K exhibit a sharp cut-off around 2 MHz for a wide range of S4max values. Similar features were also found in 13 ionosondes with lower cut-off frequencies in the range 1.28–1.60 MHz. It indicates that $f_o$Es does not vary with S4max and $f_o$Es is determined less reliably in the low frequency. It is different from the lowest frequency of reflection waves due to the physical limits of ionosondes. These ionosondes are all manually scaling stations and so it is likely that the influence of ambient ionizations within the background E layer is responsible for outlier $f_o$Es near the low-frequency scaling threshold. The horizontal red lines in figure 1 represent the scaling threshold for each ionosonde, below which the $f_o$Es is determined less reliably, identified by an abnormally high occurrence of Es layers below this frequency (figure 2). In general, the manually scaled ionosonde data are more accurate than the auto-scaled values. However, there remain some issues with manually scaled data near the scaling threshold. To avoid this problem of the ionosonde detection limits, data scalers could disregard Es layers observed within the frequency range 1.28–1.60 MHz, but this would then lead to another problem of an overestimate of $f_o$Es as discussed in the next section. In addition, errors in the manually scaled $f_o$Es data may also be caused by the instrumental change or the data being scaled by different people.

Figure 2 shows a density scatter plot of the coincident measurements from FORMOSAT-3/COSMIC satellites and ground-based ionosondes in the period 2006–2014. An established methodology for the

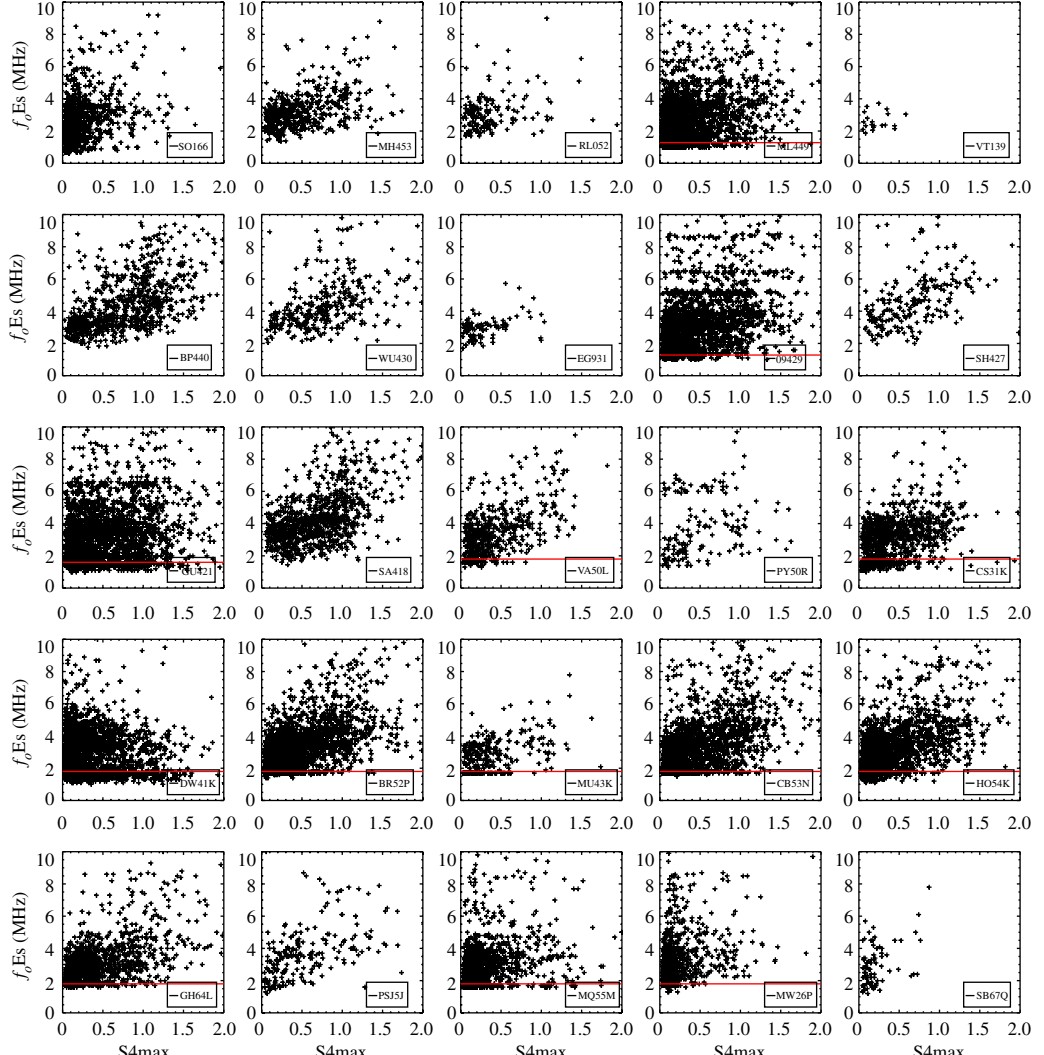

**Figure 1.** Comparison of the hourly intensity of Es layers characterized by S4max from FORMOSAT-3/COSMIC RO measurements and $f_oEs$ by ground-based measurements for the north–south (latitude) 25 ionosonde stations in the period 2006–2014. The horizontal red lines represent the scaling threshold for each ionosonde, below which the $f_oEs$ is determined less reliably. The details of 25 ionosonde stations are listed in table 1.

fitting S4max and $f_oEs$ was applied [62]. S4max is an index of an amplitude scintillation resulting from vertical gradients in the ionospheric irregularities. In previous studies, S4max was found to be linearly related to $f_oEs$ or related to the electron density of ionospheric irregularities (equivalently the square of plasma frequencies $f^2$) [53,63]. The observations were binned in 0.04 (S4max) × 0.20 MHz ($f_oEs$). Bins within the colour scale contain at least 8 coincident observations. A majority of observations (90%) have a much greater number. $f_oEs$ values were scattered over a large range from 0.5 MHz to 6.5 MHz. A correlation (correlation coefficient: $r = 0.44$) was found between S4max and $f_oEs$ for the relation $f_oEs = 2.43 + 1.75 \times$ S4max based on all the hourly coincident measurements, although this will be influenced by variations of the instrumental sensitivity. The $p$-value is less than 0.01 (for the test that two datasets are independent). The violet line represents a linear fit between S4max and $f_o^2Es$, which yields the relation $f_o^2Es = 6.13 + 14.66 \times$ S4max ($r = 0.42$, $p < 0.01$). The correlation between S4max and $f_oEs$ could be affected by the outliers in S4max and $f_oEs$ data. It could also be influenced by the local variability within Es layers in an accumulated period of 1 h and spatial variation within the assumed geographical region of 5° latitude × 5° longitude.

Another issue is that the scaled $f_oEs$ is less reliable for the observations near the scaling threshold of 1.28–1.60 MHz. In order to account for this, the analysis was restricted to $f_oEs$ above the scaling threshold for each instrument. $f_oEs$ values below the red horizontal line in figure 1 were not considered. Figure 3 shows the density scatter plot of 22 234 h coincident events after removing

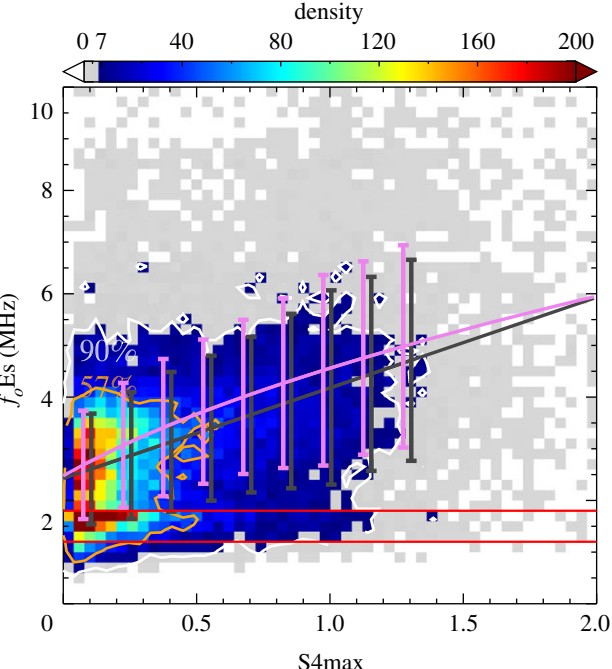

**Figure 2.** Density scatter plot of hourly $f_o$Es measured by ionosondes and the S4max from FORMOSAT-3/COSMIC in the period 2006 to 2014. The black line represents the linear least-squares fit between $f_o$Es and S4max, $f_o$Es $= 2.43 + 1.75 \times$ S4max. The violet line represents the linear least-squares fit between $f_o^2$Es and S4max represented by the equation $f_o^2$Es $= 6.13 + 14.66 \times$ S4max. The two horizontal red lines represent the lower threshold for manual scaling between 1.28 and 1.60 MHz.

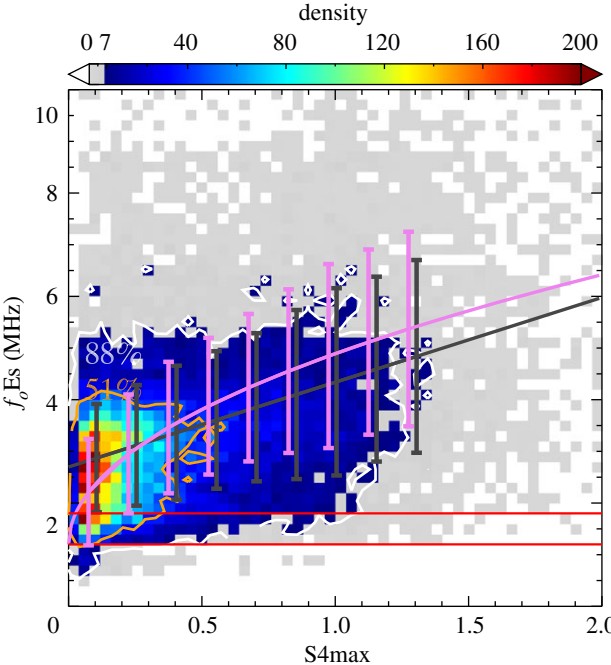

**Figure 3.** Density scatter plot of hourly $f_o$Es measured by ionosondes after removing potentially erroneous data below the scaling threshold in the manual-scaled values, and the S4max from FORMOSAT-3/COSMIC in the period 2006–2014. The black line represents the linear least-squares fit between $f_o$Es and S4max by the equation $f_o$Es $= 2.70 + 1.64 \times$ S4max. The violet line represents a fit determined by a simple formula based on the physical interpretation of S4max, $(f_o$Es $- 1.2)^2 = 13.62 \times$ S4max.

the potentially erroneous $f_o$Es data below the scaling threshold. The GPS/GNSS RO technique is a unique tool for detecting weak Es layers ($f_o$Es $< 3.6$ MHz and S4max $< 0.4$) with many more detections occurring within this range than above it [62]. Conversely, an ionosonde is most efficient at

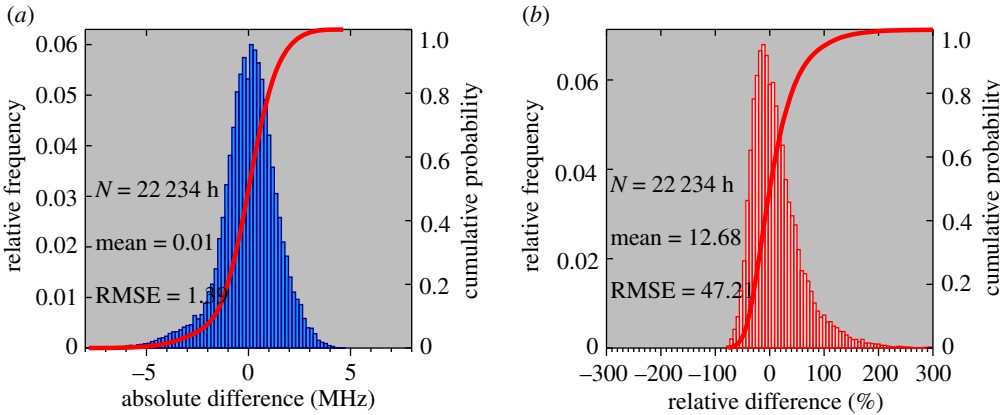

**Figure 4.** Statistical analyses of the absolute difference and relative difference between $f_oEs$ values from ionosondes and COSMIC data using the equation $(f_oEs - 1.2)^2 = 13.62 \times S4max$ for 25 stations in the period 2006 to 2014.

detecting strong Es layers. Either detection technique will only measure a subset of the distribution of the intensity of Es layers [62]. The correlation between the two techniques will be influenced by these factors. Nevertheless, a general characteristic of their relationship can be fitted by a comparatively simple formula. Since the S4max is related to the fluctuations of GPS/GNSS RO signals caused by large vertical gradients in the Es layer, the magnitude of S4max has a linear dependence on the electron density [63]. Therefore, the relationship between S4max and $f_o^2Es$ is defined as $(f_oEs - f_{BG})^2 = a \times S4max$. $f_{BG}$ is the background frequency of the ambient electron density in the absence of Es layers, which is 1.2 MHz estimated from the low detection threshold of Es layers by ionosondes. The violet curve in figure 3 represents the best-fit relation, which is $(f_oEs - 1.2)^2 = 13.62 \times S4max$ ($r = 0.40$, $p < 0.01$).

Based on the fitted curve $(f_oEs - 1.2)^2 = 13.62 \times S4max$, $f_oEs$ can now be derived for all the FORMOSAT-3/COSMIC S4max data. Figure 4a,b shows the statistical analyses of the absolute difference and relative difference between measured and derived $f_oEs$ values. The $f_oEs$ absolute difference ($f_{COSMIC} - f_{ionosonde}$) shows a typical Gaussian distribution. The mean and the root mean square error (RMSE) are 0.01 and 1.39 MHz. The relative $f_oEs$ difference (($f_{COSMIC} - f_{ionosonde}$)/ $f_{ionosonde}$) shows 4 631 out of 22 234 (20.83%) hourly coincident measurements have a relative difference less than 10%. A total of 58.50% coincident measurements have a relative difference less than 30%, and 80.61% coincident measurements have a relative difference less than 50%. The mean and RMSE of the relative difference are 12.68% and 47.21%.

The comparisons between hourly $f_oEs$ obtained by ionosondes and derived from the FORMOSAT-3/ COSMIC RO measurements for 25 stations during 2006–2014 are shown in figure 5. The scatter plots show a 'ledge' extending to the right side of the ideal fitting line $y = x$, in which the $x$-axis $f_oEs$ was produced by ionosonde scaling processing when the $y$-axis $f_oEs$ was derived from the COSMIC S4max based on the relationship between the $f_oEs$ and S4max. Since some small values of $f_oEs$ less than 1.28–1.60 MHz cannot be well identified by the ground-based ionosondes, the derived $f_oEs$ from the FORMOSAT-3/COSMIC S4max was slightly underestimated from 5 to 10 MHz. Besides, only a few coincident measurements of the Es layer have values of $f_oEs$ exceeding 6 MHz. A few large S4max values were not observed because of the occurrence of interruptions in GNSS signals when the plasma frequency $f_oEs$ exceeds approximately 6 MHz.

Figure 6 shows the daily $f_oEs$ smoothed with a 5-day running mean for measured (ionosondes, black) and derived (FORMOSAT-3/COSMIC, yellow) values for 25 stations in the period 2006 to 2014. The climatological variability and local perturbations in the Es layer derived from the FORMOSAT-3/ COSMIC RO signals agree with the ionosonde observations. The FORMOSAT-3/COSMIC $f_oEs$ therefore provides an important measure of Es layers complementary to ground-based stations, particularly in filling in the gaps in ionosonde data over a specific period of time or over the region with no coverage from ionosonde stations. $f_oEs$ values appear to be overestimated for some ionosondes (WU430, SH427 and SA418) which may be due to decisions made during manual scaling to avoid the erroneous data close to the threshold of 1.28–1.60 MHz. In addition, the occurrence of multiple echoes at higher altitudes may also explain this effect [64]. Ionospheric plasma stratifications in the ionosonde data can be more easily identified by the manual scaling.

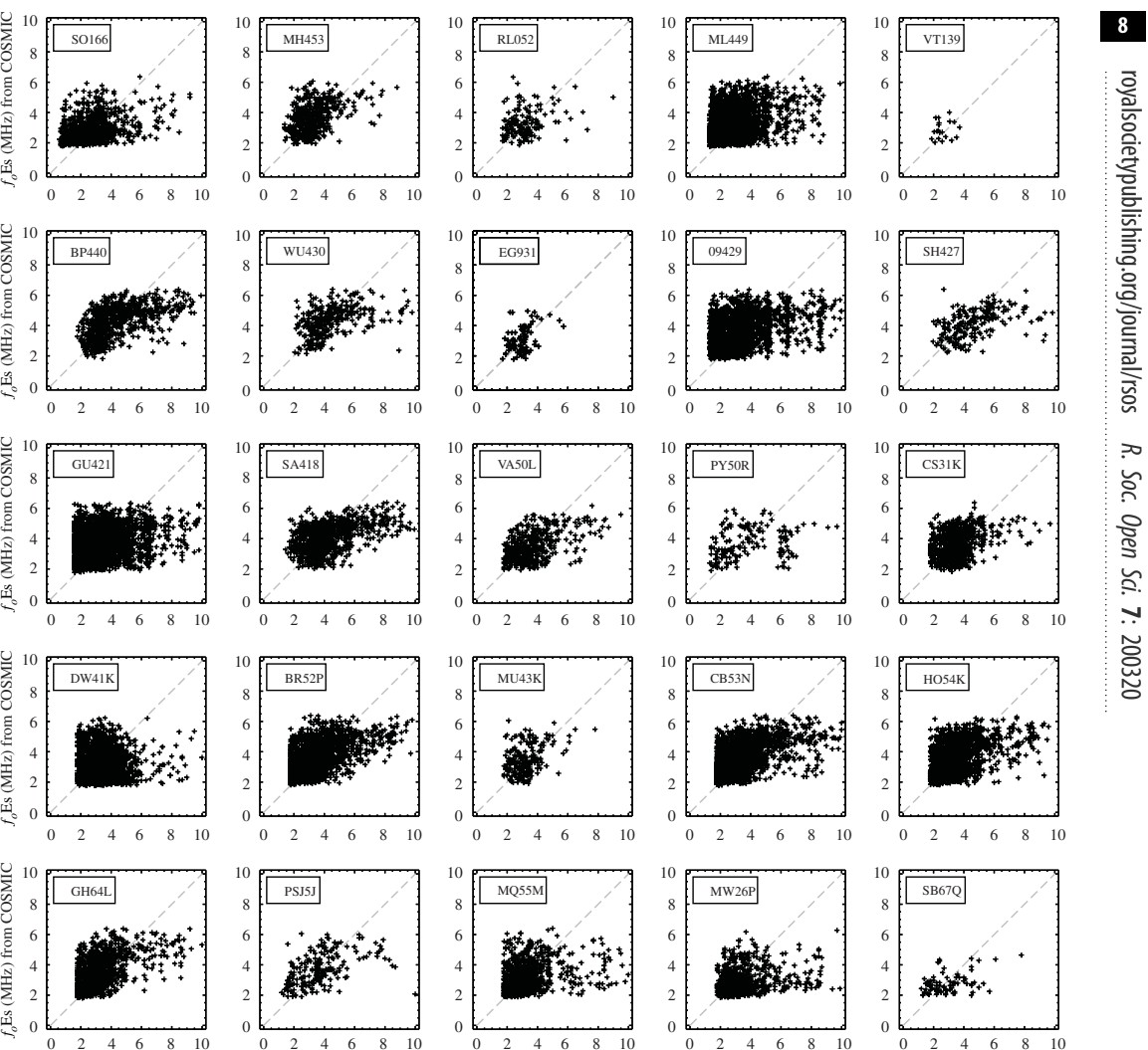

**Figure 5.** Comparison between the hourly $f_o$Es (MHz) obtained by ionosondes and derived from the FORMOSAT-3/COSMIC RO measurements using the equation $(f_o\text{Es} - 1.2)^2 = 13.62 \times \text{S4max}$ for 25 stations in the period 2006 to 2014.

## 4. Results

Figure 7 shows the global map of $f_o$Es derived from S4max data in a $1° \times 1°$ grid in the period 2006–2014. The geomagnetic latitude contours of $60°$, $70°$ and $80°$ are plotted as red lines in the Northern Hemisphere and green lines in the Southern Hemisphere. The geomagnetic equator is presented as a yellow line. The yellow dots show the distribution of 2638 FORMOSAT-3/COSMIC RO events in 24 h on 25 December 2007. An intense Es layer with average $f_o$Es exceeding 3.5 MHz occurs at mid-latitudes ($5°$–$60°$) and high latitudes ($80°$–$90°$). Weaker Es layers with $f_o$Es around 2.5 MHz form troughs that are visible near the magnetic equator and along the $60°$–$80°$ geomagnetic latitude bands. The dependence of the distribution of $f_o$Es on the geomagnetic field can be explained by the effects of Lorentz forces on the Es plasma during wind shear action [16]. Auroral Es layers at high latitudes were long assumed to be a direct manifestation of the aurora. However, in fact, the high-latitude Es layers are much thinner than those produced by auroral precipitation. Wind shear drift convergence and ion vertical transport by the electric field could be responsible for these Es layers at geomagnetic high latitudes [65,66]. The derived $f_o$Es provides a standard reference to assess the quality of observations of $f_o$Es recorded by different types of ionosondes. Though manually scaled $f_o$Es values are more reliable than autoscaled values, changes in instrumentation and scaling criteria for Es by different people may have an impact on the accuracy of the manual-scaling process. In figure 7, the circles represent the location of 25 ionosondes used in the analysis. The correlation coefficients

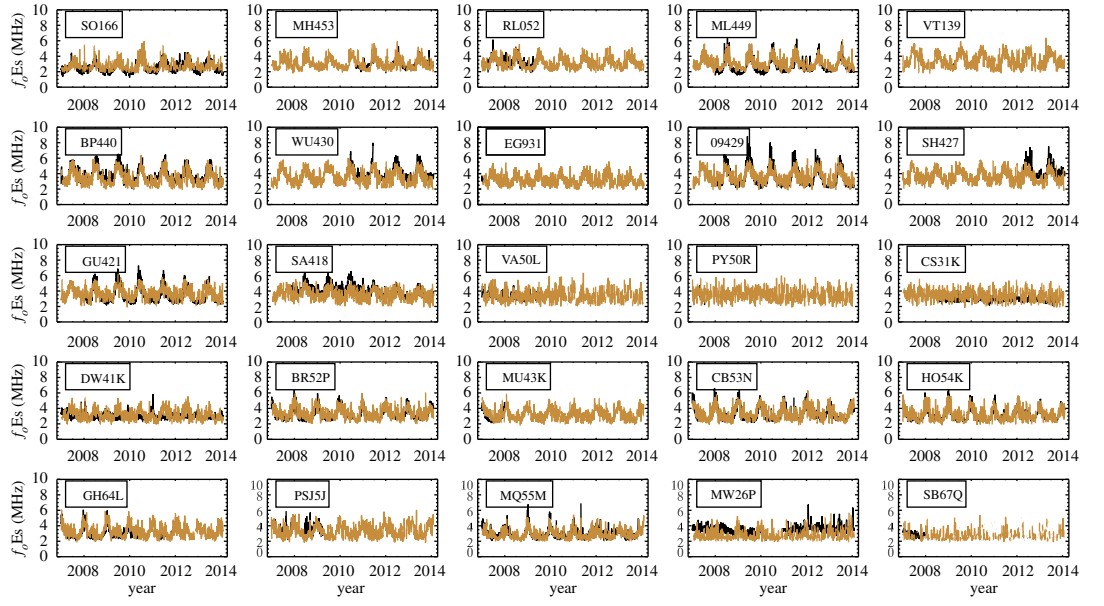

**Figure 6.** Time series of the 5-day-smoothed daily mean $f_o$Es from ionosondes (measured, black) and FORMOSAT-3/COSMIC Satellites (derived, yellow) for 25 stations in the period 2006 to 2014.

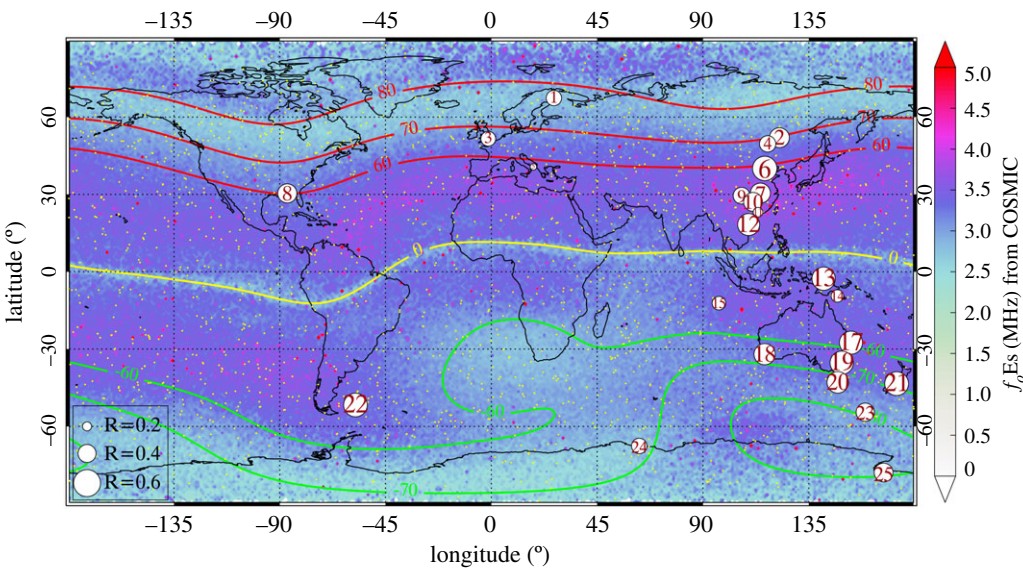

**Figure 7.** Global geographical distribution of $f_o$Es derived from FORMOSAT-3/COSMIC S4max in 2006–2014 in a 1° × 1° grid. The red and green lines represent the geomagnetic latitude contours of 60°, 70° and 80° in the Northern Hemisphere and Southern Hemisphere. The yellow line represents the geomagnetic equator. The locations of 25 ionosondes in the analysis are shown as circles with the correlation coefficient between derived $f_o$Es and $f_o$Es by individual ionosondes represented by the size of symbols. The yellow dots show the distribution of 2638 FORMOSAT-3/COSMIC RO events in 24 h on 25 December 2007.

between $f_o$Es derived from FORMOSAT-3/COSMIC and $f_o$Es observed by individual ionosondes are denoted by the size of these symbols. Fifteen out of 25 stations (MH453, BP440, WU430, EG931, SH427, SA418, VA50L, BR52P, MU43K, CB53N, HO54K, GH64L, PSJ5J, MQ55M and SB67Q) have a correlation between 0.4 and 0.6. Except for the data from PSJ5J, all other values of $f_o$Es are manual-scaled observations.

A plot of $f_o$Es versus geographical latitude is presented in figure 8. The mean $f_o$Es values derived from FORMOSAT-3/COSMIC are shown as a red line with the standard deviation from the mean within a 1° latitude band represented by the width of the pink shaded area about this line. The blue dots represent

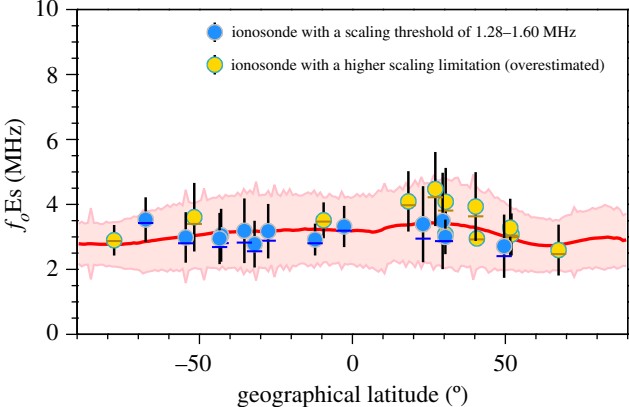

**Figure 8.** Mean $f_o$Es values observed by the individual ionosonde stations (blue and yellow dots for mean, dashes for median, and up-down bars for standard deviation) and FORMOSAT-3/COSMIC (red line) versus geographical latitude in the period 2006–2014. The standard deviations of FORMOSAT-3/COSMIC-derived $f_o$Es are represented by the width of the pink shaded area.

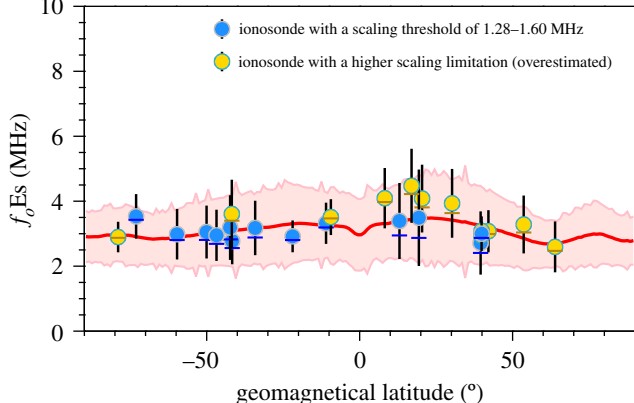

**Figure 9.** Mean $f_o$Es values by the individual ionosonde stations (blue and yellow dots for mean, dashes for median, and up-down bars for standard deviation) and FORMOSAT-3/COSMIC (red line) versus geomagnetic latitude in the period 2006 to 2014. The standard deviations of FORMOSAT-3/COSMIC-derived $f_o$Es are represented by the width of the pink shaded area.

$f_o$Es observed by ionosondes with a scaling threshold of 1.28–1.60 MHz (ML449, 09429, GU421, VA50L, CS31K, DW41K, BR52P, MU43K, CB53N, HO54K, GH64L, MQ55M and MW26P). The yellow dots represent $f_o$Es observed by ionosondes with a higher scaling limitation (SO166, MH453, RL052, VT139, BP440, WU430, EG931, SH427, SA418, PY50R, PSJ5J and SB67Q), which introduces a bias toward an overestimation of $f_o$Es. This implies that while raising the lower threshold avoids using these potentially erroneous data, the distribution of the measured $f_o$Es will be biased towards larger values, resulting in an overestimate of $f_o$Es.

Figure 9 shows the plot of $f_o$Es versus the geomagnetic latitude. The intensity of the Es layer is geomagnetically controlled. The dependence of $f_o$Es on the geomagnetic latitude is stronger than that on the geographic latitude. The Es layer is slightly weaker in the equatorial region and high northern geomagnetic latitudes of 60° N–70° N. A slightly more pronounced summer peak in $f_o$Es occurs at mid-latitudes between 10° and 30°. A secondary peak of $f_o$Es occurs at 80° geomagnetic latitude probably resulting from the gravity waves, tidal winds and electric fields at high latitudes [20,65]. The distribution of these derived $f_o$Es is therefore consistent with the observed occurrence of Es layers [23].

The high-sensitivity and global measurements of the FORMOSAT-3/COSMIC RO technique provide an opportunity to investigate the seasonal behaviour of Es layers through variations in $f_o$Es. Figure 10 shows the altitude–local time distribution of $f_o$Es for four seasons in different latitudinal regions in the Northern Hemisphere in the period 2006–2014. In summer, $f_o$Es can reach 4.5–5.0 MHz at mid-latitudes between 15°N and 45°N within the altitude range 100–130 km. In winter, the Es layer has a

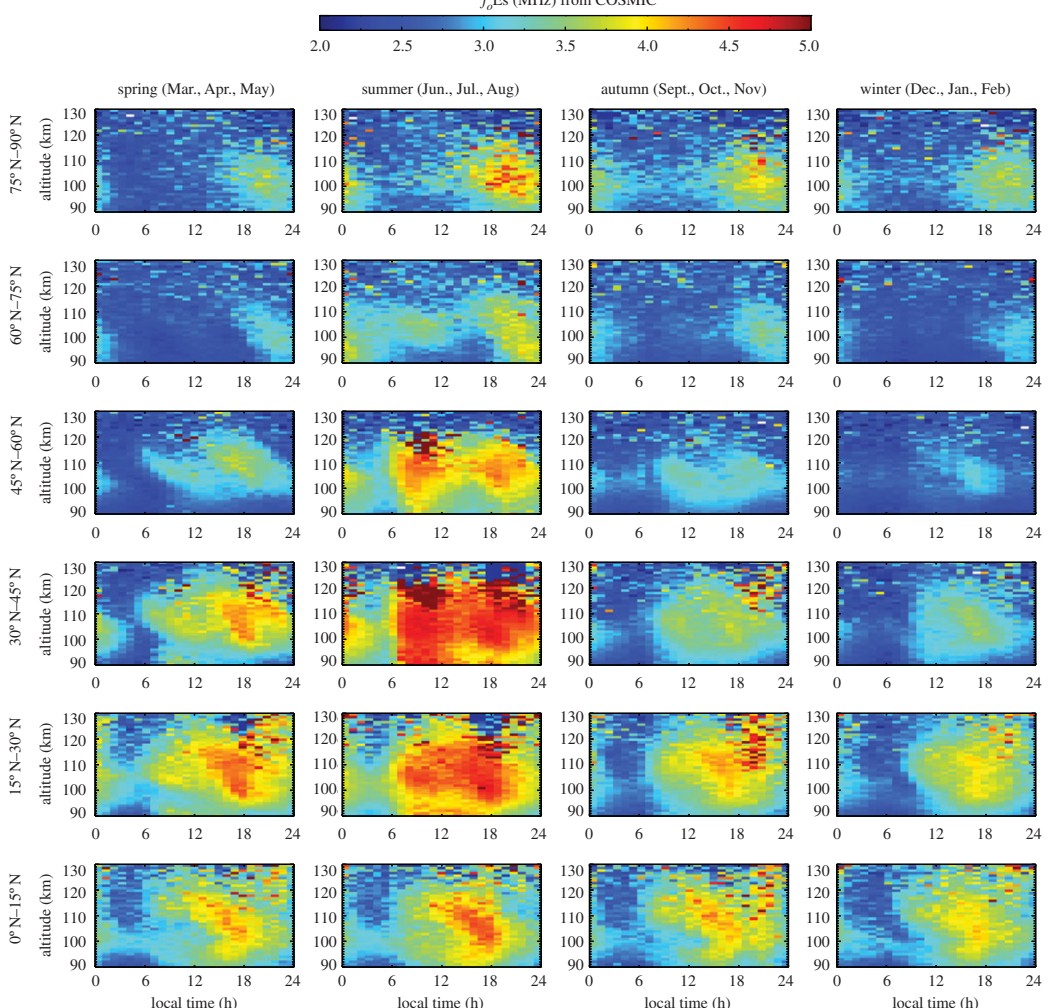

**Figure 10.** Seasonal variation in the altitude–ocal time distribution of $f_o$Es in different latitudes in the Northern Hemisphere in the period 2006–2014.

minimum $f_o$Es value of 2.0–2.5 MHz and a maximum $f_o$Es value of 3.5–4.0 MHz. Figure 11 shows the altitude–local time distribution of $f_o$Es in the Southern Hemisphere. The summer maximum of $f_o$Es is 4.0–4.5 MHz at 15° S–60° S latitude in the altitude range 100–125 km. In winter, the minimum $f_o$Es is 2.0–2.5 MHz and the maximum $f_o$Es is 3.5–4.0 MHz.

The seasonal differences in $f_o$Es are due to the dynamics in Es layers, which are directly affected by the wind shear convergence nodes descending with diurnal and semidiurnal tides [24]. Es layers are often referred to as 'tidal ion layers' since the atmospheric tides play a fundamental role in the formation and the descent of these layers [22]. The diurnal and semidiurnal tides control the descent of Es from 120 km down to 100 km. The influence of the semidiurnal tide is prevalent in June and July, and the influence of the diurnal tide is prevalent in September [67]. The transition from the diurnal tide at low latitudes to the semidiurnal tide at high latitudes occurs at 40° latitude [68]. The tidal variability in Es layers has been studied by the ISR and ionosonde [22,24]. The global maps of $f_o$Es in figures 10 and 11 verify the tidal variations in Es layers and the resulting altitude descent of Es. In figure 10, a semidiurnal pattern in $f_o$Es dominates at mid-latitudes between 30° N and 75° N in spring and summer with a descent speed of 1.6–2.5 km h$^{-1}$. The Es layer descends from approximately 120 km at 06.00 and 18.00 local time, which agrees with the variations in the occurrence rate of Es layers [52]. The diurnal pattern in $f_o$Es dominates at low latitudes between 0–30° N. In the Southern Hemisphere (figure 11) likewise, the pattern of a diurnal periodicity in the Es layer occurs at low latitudes between 0 and 30° S, and the pattern of a semidiurnal periodicity occurs at mid-latitudes between 30° S and 60° S in spring and summer.

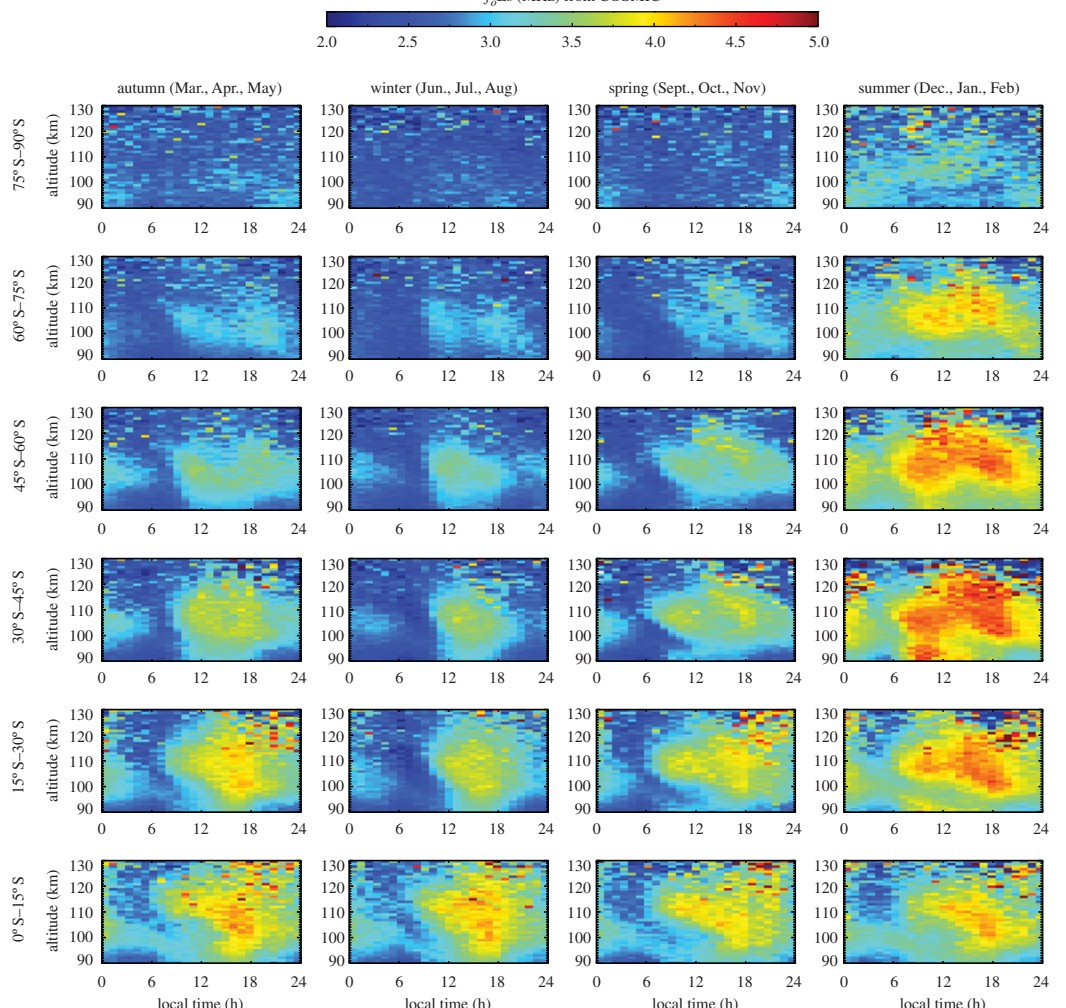

**Figure 11.** Seasonal variation in the altitude–local time distribution of $f_oEs$ in different latitudes in the Southern Hemisphere in the period 2006–2014.

## 5. Conclusion

Irregularities in electron number density within Es layers between 90 and 130 km altitude can significantly influence GPS/GNSS signals. Approximately 23% of GNSS signals from LEO-based receivers are interrupted or even lost [42]. Over one-third of ionospheric perturbations responsible for the transient loss of lock in the GNSS receiver tracking loops were caused by ionospheric Es layers. Fluctuations in these GNSS signals can be used to investigate the global Es layers.

In this study, global hourly $f_oEs$ data were derived based on the FORMOSAT-3/COSMIC RO measurements correlated with ground-based ionospheric data. A total of 5.8 million S4max observations occurring between altitudes of 90–130 km from the FORMOSAT-3/COSMIC RO signals and observations of Es layers by 25 ionosonde stations were used. The high-resolution and high-sensitivity RO technique makes it possible to determine the global distribution of Es layers at a high spatial and temporal resolution, even weak Es layers that are below the threshold of reliable detection for ground-based ionosondes.

The global distribution of $f_oEs$ in a 1° × 1° grid shows a strong dependence on the geomagnetic latitude. The mean $f_oEs$ values exceeding 3.5 MHz are predominantly distributed at geomagnetic latitudes of 5°–70° and 80°–90°. $f_oEs$ is approximately 2.5 MHz near the magnetic equator and along the 70°–80° geomagnetic latitude bands. The summer maximum $f_oEs$ in the Northern Hemisphere is 4.5–5.0 MHz which occurs at mid-latitudes between 15° N and 45° N at 100–130 km. The summer maximum $f_oEs$ in the Southern Hemisphere is 4.0–4.5 MHz which occurs at mid-latitudes between 15° S and 60° S at 100–120 km. In winter, the maximum $f_oEs$ is around 4.0 MHz at 100–120 km, which

occurs at low latitudes between 0° N and 30° N and 0°S–15° S. The winter minimum is approximately 2.0–2.5 MHz. The global distribution of $f_oEs$ verifies previous studies of tidal variations in Es layers and the altitude descent with local time observed by several independent ground-based monitoring stations [22,24,67,68]. The diurnal tide dominates low latitudes between 0–30° and high latitudes between 75° and 90°. The semidiurnal tide dominates at mid-latitudes between 30° and 75° in spring and summer.

The follow-on constellation FORMOSAT-7/COSMIC-2 launched on 25 June 2019, and it is expected to provide 3–4 times the amount of high-quality RO profiles as the previous FORMOSAT-3/COSMIC satelliltes [69]. A dramatically increased number of RO observations will enable observational investigations of the short-term variability in Es layers and may eventually improve the capability of ionospheric forecasts, which will benefit the applications of GPS/GNSS precise-point positioning. The Martian Es layer was recently discovered by the NASA's Mars Atmosphere and Volatile Evolution spacecraft [70]. It highlights the role of the planetary Es layers in long-distance radio communications for planetary exploration.

Data accessibility. The global critical frequency $f_oEs$ data derived from the FORMOSAT-3/COSMIC S4max in the period 2006 to 2014 in this work have been deposited at Dryad: doi:10.5061/dryad.xsj3tx9bx. The COSMIC S4max data are available from the CDAAC website (https://cdaac-www.cosmic.ucar.edu/cdaac/). The ionosonde data are available from the Data Centre for Meridian Space Weather Monitoring Project (https://data.meridianproject.ac.cn), the Institute of Geology and Geophysics, Chinese Academy of Sciences (http://space.iggcas.ac.cn), and the UKSSDC at the Rutherford Appleton Laboratory (https://www.ukssdc.ac.uk).

Authors' contributions. B.Y. designed the experiments, B.Y. and C.J.S. performed data analysis and wrote the manuscript. X.X. and X.Y. provided the FORMOSAT-3/COSMIC radio occultation data. X.X. and X.D. provided the ionosonde data in China. X.X., X.Y. and X.D. contributed to the discussion of the results. All authors gave final approval for publication.

Competing interests. The authors declare no competing interests.

Funding. We thank the Royal Society for the Newton International Fellowship of Bingkun Yu for supporting his research. This work is also supported by the National Natural Science Foundation of China (grant nos. 41774158, 41974174 and 41831071), B-type Strategic Priority Program of CAS grant no. XDB41000000, and the Fundamental Research Fund for the Central Universities.

Acknowledgements. We acknowledge the FORMOSAT-3 (FORMosa SATellite Mission-3)/COSMIC (Constellation Observing System for Meteorology, Ionosphere and Climate) radio occultation data, the ionosonde data from the UK Solar System Data Centre (UKSSDC) at the Rutherford Appleton Laboratory, the Chinese Meridian Project, the Solar-Terrestrial Environment Research Network (STERN), the Data Center for Geophysics, Data Sharing Infrastructure of Earth System Science, and the National Science & Technology Infrastructure of China.

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
