## [Reviewer comments · Royal Society Open Science]

Review History

RSOS-200320.R0 (Original submission)

Review form: Reviewer 1

Is the manuscript scientifically sound in its present form?

No

Are the interpretations and conclusions justified by the results?

No

Is the language acceptable?

No

Do you have any ethical concerns with this paper?

No

Have you any concerns about statistical analyses in this paper?

No

Recommendation?

Major revision is needed (please make suggestions in comments)

Comments to the Author(s)

This paper describes a method for obtaining sporadic E critical frequencies from satellite GPS radio occultation measurements. This is achieved by using a large network of ground-based ionosondes to correlate with FORMOSAT-3/COSMIC satellite measurements. The authors quickly discover that the correlations (Fig. 1) are not very strong, and do some work to deal with ionosonde measurements close to the instrumental detection limits. An improved correlation is derived, and this will be of value to people using space-based radio occultation measurements to study sporadic E layers.

The paper contains many grammatical errors and missing references indicated in the text by [?]. In fact, I have rarely been asked to review a paper that has so clearly not been carefully proof-read before submission. Nevertheless, most of it - apart from in a few places - is perfectly understandable. I have three major concerns:

1. Does any new science come out of this study? All the conclusions about geomagnetic latitudinal dependence and tidally-driven layer descent are already well known (and cited in the text). So are there any discoveries that have come out of this study?
2. I am not sure that the data really supports your conclusions. For example, the statement on page 7 "The troughs of Es layers with foEs around 2.5 MHz were visible near the magnetic equator and along the 60 – 80 geomagnetic latitude bands" is not borne out by Figure 8. This shows - within the error band - no latitudinal dependence. The same point applies to the statement at the top of page 8.
3. A second example concerns the apparent tidal descent discussed on page 8 around line 20. Many of the plots in Figure 9 do not show - from visual inspection - any evidence of descent, either diurnal or semi-diurnal. So you need to explain why at some latitudes/seasons there is evidence of descent, but not in others, or providing a more easily understood and convincing analysis.

Minor points/corrections (this is only a small sub-set - please get a native English speaker to proof read the paper - it should NOT be the job of the reviewers):

page 4: need to define what the critical frequency and S_4^{\max} are in the introduction, not later on in the paper.

page 4: "Whereas, it remains one weak point in explaining the seasonal dependence of Es that is the geographical distribution of Es minimum in the winter hemisphere cannot be simulated by the neutral wind shears.." So, having mentioned this problem, do you solve it in this study?

page 5: "Besides, the performance of ground-based observations from the world-wide ionosonde network can be evaluated by the comparative results." What does "comparative" mean here - comparing with other ionosondes?

page 5: "Six FORMOSAT-3/COSMIC satellites initially followed the same orbit at ~512 km and then sequentially orbit the Earth at ~800 km." This is unclear, what is meant by "sequentially"?

page 5, line 36: some problem with characters "1500 \tilde{A} ,S2500"

page 5: "The ground-truth observations are almost "manually scaled or automatic edited values" - unclear

page 6, line 14: "Thses ionosondes"

page 6, line 47: "the general relationship is loose to be well derived" What does this mean?

page 7, line 37: "The high energy radiation, particle precipitation, and polar gap gravity waves could be responsible for the Es layer at geomagnetic high latitudes." What "high energy radiation" would only affect the atmosphere above 80 degrees geomagnetic? Auroral precipitation occurs in the polar cap, which is at lower geomagnetic latitudes.

page 7, line 50: "the standard deviation as the error-in-the-mean" - what else is a standard deviation?

Review form: Reviewer 2

Is the manuscript scientifically sound in its present form?

Yes

Are the interpretations and conclusions justified by the results?

Yes

Is the language acceptable?

Yes

Do you have any ethical concerns with this paper?

No

Have you any concerns about statistical analyses in this paper?

No

Recommendation?

Major revision is needed (please make suggestions in comments)

Comments to the Author(s)

Here are some of my comments and suggestion for improvement of the paper.

It has not yet been clearly proven that the lightning has relation to Es layer formation. Some theories and their statistical supports were proposed and prepared. However, the significant issues of those works have been addressed in works of Haldoupis.

Check misspellings pg.6 ln.15,

Incomplete references pg.5 ln. 40, 41

Missing reference pg.6 ln. 26, 33, 44, 50, pg.8 ln. 12, pg.7. ln. 8, 12, pg.9 ln. 9, 12

Missing reference pg.8, ln. 12, pg.7. ln.8, 12

Stations names are difficult to read Fig.1, where the latitude could be found on a particular plot?

How the correlation can be proved, what type of correlation/pattern has been found. According to me, there no clear relationship seen. There is no supporting analysis provided for statement:

A general correlation is found between foEs by the ionosondes and FORMOSAT-3/COSMIC S₄max at different latitudes.

I would like to point out the fact that problems in scaling foEs close to lower limit of the sounding range is rather related to technical problems due to physical conditions than to the fact that foEs is manually scaled. The physical limits of sounding close to lower ionosonde limit should be discussed instead. Same physical problems affect automatically scaled data too.

What is the physical meaning of the fitting curves (Fig.2, Fig.3)? How could the use of the linear fitting tell us about the Es layer? And particularly, what would be the use of the interpretation at S4max for values 1.5 or 2? Basically, any fitting can be use any time and some results are obtained. However, there should be some physical background or meaning for type of the chosen fitting.

Interpretation of Fig. 5

foEs values appear to be overestimated by some ionosondes (WU430, SH427 and SA418) as a result of raising a scaling limitation of frequency to avoid the erroneous data close to the threshold of 1.28–1.60 MHz.

Within the text there is no information about multiple layer stratification which may explain the effect as well. It clearly shows the importance of manual scaling where one can easily see the fine structure and distinguish particular layer stratification.

Correct typos!

Decision letter (RSOS-200320.R0)

Dear Dr Yu,

The editors assigned to your paper ("Derivation of global ionospheric Sporadic E critical frequency (foEs) data from the amplitude variations in GPS/GNSS radio occultations") have now received comments from reviewers. We would like you to revise your paper in accordance with the referee and Associate Editor suggestions which can be found below (not including confidential reports to the Editor). Please note this decision does not guarantee eventual acceptance.

The Editors ask that you seek English language editing advice, either from a colleague or possibly via <https://royalsociety.org/journals/authors/benefits/language-editing/>.

Please submit a copy of your revised paper before 15-May-2020. Please note that the revision deadline will expire at 00.00am on this date. If we do not hear from you within this time then it will be assumed that the paper has been withdrawn. In exceptional circumstances, extensions may be possible if agreed with the Editorial Office in advance. We do not allow multiple rounds of revision so we urge you to make every effort to fully address all of the comments at this stage. If deemed necessary by the Editors, your manuscript will be sent back to one or more of the original reviewers for assessment. If the original reviewers are not available, we may invite new reviewers.

When submitting your revised manuscript, you must respond to the comments made by the referees and upload a file "Response to Referees" in "Section 6 - File Upload". Please use this to document how you have responded to the comments, and the adjustments you have made. In

order to expedite the processing of the revised manuscript, please be as specific as possible in your response.

- Data accessibility

If you wish to submit your supporting data or code to Dryad (<http://datadryad.org/>), or modify your current submission to dryad, please use the following link:
<http://datadryad.org/submit?journalID=RSOS&manu=RSOS-200320>

- Competing interests

- Authors' contributions

- Acknowledgements

- Funding statement

Once again, thank you for submitting your manuscript to Royal Society Open Science and I look

forward to receiving your revision. If you have any questions at all, please do not hesitate to get in touch.

on behalf of Dr Cécile Lasserre (Associate Editor)
 openscience@royalsociety.org

Associate Editor's comments (Dr Cécile Lasserre):

Associate Editor: 1

Comments to the Author:

Both reviewers agree on the scientific interest of the paper but major revisions are required before publication. Please answer reviewers' comments and suggestions in details in your revised version. Make sure to better highlight the novelty of your paper with respect to already published work and conclusions, especially on the methodological aspects, and to strengthen data analysis, as advised.

The level of language must also be improved, with the help of the native-english co-author.

Comments to Author:

Reviewers' Comments to Author:

Reviewer: 1

Comments to the Author(s)

This paper describes a method for obtaining sporadic E critical frequencies from satellite GPS radio occultation measurements. This is achieved by using a large network of ground-based ionosondes to correlate with FORMOSAT-3/COSMIC satellite measurements. The authors quickly discover that the correlations (Fig. 1) are not very strong, and do some work to deal with ionosonde measurements close to the instrumental detection limits. An improved correlation is derived, and this will be of value to people using space-based radio occultation measurements to study sporadic E layers.

The paper contains many grammatical errors and missing references indicated in the text by [?]. In fact, I have rarely been asked to review a paper that has so clearly not been carefully proof-read before submission. Nevertheless, most of it - apart from in a few places - is perfectly understandable. I have three major concerns:

1. Does any new science come out of this study? All the conclusions about geomagnetic latitudinal dependence and tidally-driven layer descent are already well known (and cited in the text). So are there any discoveries that have come out of this study?
2. I am not sure that the data really supports your conclusions. For example, the statement on page 7 "The troughs of Es layers with foEs around 2.5 MHz were visible near the magnetic equator and along the 60 – 80 geomagnetic latitude bands" is not borne out by Figure 8. This shows - within the error band - no latitudinal dependence. The same point applies to the statement at the top of page 8.
3. A second example concerns the apparent tidal descent discussed on page 8 around line 20. Many of the plots in Figure 9 do not show - from visual inspection - any evidence of descent, either diurnal or semi-diurnal. So you need to explain why at some latitudes/seasons there is

evidence of descent, but not in others, or providing a more easily understood and convincing analysis.

Minor points/corrections (this is only a small sub-set - please get a native English speaker to proof read the paper - it should NOT be the job of the reviewers):

page 4: need to define what the critical frequency and S_4 max are in the introduction, not later on in the paper.

page 4: "Whereas, it remains one weak point in explaining the seasonal dependence of Es that is the geographical distribution of Es minimum in the winter hemisphere cannot be simulated by the neutral wind shears.." So, having mentioned this problem, do you solve it in this study?

page 5: "Besides, the performance of ground-based observations from the world-wide ionosonde network can be evaluated by the comparative results." What does "comparative" mean here - comparing with other ionosondes?

page 5: "Six FORMOSAT-3/COSMIC satellites initially followed the same orbit at ~512 km and then sequentially orbit the Earth at ~800 km." This is unclear, what is meant by "sequentially"?

page 5, line 36: some problem with characters "1500 \hat{a} A ,S2500"

page 5: "The ground-truth observations are almost manually scaled or automatic edited values" - unclear

page 6, line 14: "Thses ionosondes"

page 6, line 47: "the general relationship is loose to be well derived" What does this mean?

page 7, line 37: "The high energy radiation, particle precipitation, and polar gap gravity waves could

be responsible for the Es layer at geomagnetic high latitudes." What "high energy radiation" would only affect the atmosphere above 80 degrees geomagnetic? Auroral precipitation occurs in the polar cap, which is at lower geomagnetic latitudes.

page 7, line 50: "the standard deviation as the error-in-the-mean" - what else is a standard deviation?

Reviewer: 2

Comments to the Author(s)

Here are some of my comments and suggestion for improvement of the paper.

It has not yet been clearly proven that the lightning has relation to Es layer formation. Some theories and their statistical supports were proposed and prepared. However, the significant issues of those works have been addressed in works of Haldoupis.

Check misstypings pg.6 ln.15,

Incomplete references pg.5 ln. 40, 41

Missing reference pg.6 ln. 26, 33, 44, 50, pg.8 ln. 12, pg.7. ln. 8, 12, pg.9 ln. 9, 12

Missing reference pg.8, ln. 12, pg.7. ln.8, 12

Stations names are difficult to read Fig.1, where the latitude could be found on a particular plot? How the correlation can be proved, what type of correlation/pattern has been found. According to me, there no clear relationship seen. There is no supporting analysis provided for statement:

A general correlation is found between foEs by the ionosondes and FORMOSAT-3/COSMIC S4max at different latitudes.

I would like to point out the fact that problems in scaling foEs close to lower limit of the sounding range is rather related to technical problems due to physical conditions than to the fact that foEs is manually scaled. The physical limits of sounding close to lower ionosonde limit should be discussed instead. Same physical problems affect automatically scaled data too.

What is the physical meaning of the fitting curves (Fig.2, Fig.3)? How could the use of the linear fitting tell us about the Es layer? And particularly, what would be the use of the interpretation at S4max for values 1.5 or 2? Basically, any fitting can be use any time and some results are obtained. However, there should be some physical background or meaning for type of the chosen fitting.

Interpretation of Fig. 5

foEs values appear to be overestimated by some ionosondes (WU430, SH427 and SA418) as a result of raising a scaling limitation of frequency to avoid the erroneous data close to the threshold of 1.28-1.60 MHz.

Within the text there is no information about multiple layer stratification which may explain the effect as well. It clearly shows the importance of manual scaling where one can easily see the fine structure and distinguish particular layer stratification.

Correct typos!

Author's Response to Decision Letter for (RSOS-200320.R0)

See Appendix A.

RSOS-200320.R1 (Revision)

Review form: Reviewer 1

Is the manuscript scientifically sound in its present form?

Yes

Are the interpretations and conclusions justified by the results?

Yes

Is the language acceptable?

Yes

Do you have any ethical concerns with this paper?

No

Have you any concerns about statistical analyses in this paper?

No

Recommendation?

Accept as is

Comments to the Author(s)

The revised version of the paper is much improved. The points that I raised have been dealt with satisfactorily.

Review form: Reviewer 2**Is the manuscript scientifically sound in its present form?**

No

Are the interpretations and conclusions justified by the results?

No

Is the language acceptable?

Yes

Do you have any ethical concerns with this paper?

No

Have you any concerns about statistical analyses in this paper?

Yes

Recommendation?

Major revision is needed (please make suggestions in comments)

Comments to the Author(s)

I suggest the authors to return to the all previous comments and read them carefully and after that to make the correction again.

Despite the efforts the authors gave to the work, I do not find it acceptable. It seems that a lot of the suggestions were not answered at all. I suggest the authors to return back to the comments and read them carefully again.

For instance, the limitations of the signal detection by ionosonds has a physical reason, that sounding takes place very close to the gyrofrequency that is given by the geomagnetic field at the specific location. This number varies according to geomagnetic coordinates of the station and is an important part of the particular sounding conditions. The corrections within the text done by the authors are just showing misunderstanding of the physics behind.

I do not find satisfying the answer why and how the interpretations are done.

I do suggest the authors to look at the paper Haldoupis, JASTP, Vol: 172, Pg. 117-121, 2018.

Decision letter (RSOS-200320.R1)

Dear Dr Yu:

On behalf of the Editors, I am pleased to inform you that your Manuscript RSOS-200320.R1 entitled "Derivation of global ionospheric Sporadic E critical frequency (foEs) data from the amplitude variations in GPS/GNSS radio occultations" has been accepted for publication in Royal Society Open Science subject to minor revision in accordance with the referee suggestions. Please find the referees' comments at the end of this email.

The reviewers and Subject Editor have recommended publication, but also suggest some minor revisions to your manuscript. Therefore, I invite you to respond to the comments and revise your manuscript.

- Ethics statement

- Data accessibility

If you wish to submit your supporting data or code to Dryad (<http://datadryad.org/>), or modify your current submission to dryad, please use the following link:
<http://datadryad.org/submit?journalID=RSOS&manu=RSOS-200320.R1>

- Competing interests

- Authors' contributions

- Acknowledgements

- Funding statement

Because the schedule for publication is very tight, it is a condition of publication that you submit the revised version of your manuscript before 03-Jul-2020. Please note that the revision deadline will expire at 00.00am on this date. If you do not think you will be able to meet this date please let me know immediately.

Kind regards,
Andrew Dunn
Royal Society Open Science Editorial Office

on behalf of Dr Cécile Lasserre (Associate Editor)
openscience@royalsociety.org

Associate Editor Comments to Author (Dr Cécile Lasserre):

Associate Editor: 1

Comments to the Author:

The revised manuscript has been much improved. One of the reviewer remains unsatisfied with your answers, in particular on what concerns the discussion on the physical mechanisms behind some of your observations or limiting their accuracy and interpretation. Although partly out of scope of your paper, I strongly encourage you to go through the first review again and improve this discussion based on the literature mentioned by this reviewer.

Associate Editor: 2

Comments to the Author:

(There are no comments.)

Reviewer comments to Author:

Reviewer: 1

Comments to the Author(s)

The revised version of the paper is much improved. The points that I raised have been dealt with satisfactorily.

Reviewer: 2

Comments to the Author(s)

I suggest the authors to return to the all previous comments and read them carefully and after that to make the correction again.

Despite the efforts the authors gave to the work, I do not find it acceptable. It seems that a lot of the suggestions were not answered at all. I suggest the authors to return back to the comments and read them carefully again.

For instance, the limitations of the signal detection by ionosonds has a physical reason, that sounding takes place very close to the gyrofrequency that is given by the geomagnetic field at the specific location. This number varies according to geomagnetic coordinates of the station and is an important part of the particular sounding conditions. The corrections within the text done by the authors are just showing misunderstanding of the physics behind.

I do not find satisfying the answer why and how the interpretations are done.

I do suggest the authors to look at the paper Haldoupis, JASTP, Vol: 172, Pg. 117-121, 2018.

Author's Response to Decision Letter for (RSOS-200320.R1)

See Appendix B.

Decision letter (RSOS-200320.R2)

Dear Dr Yu,

It is a pleasure to accept your manuscript entitled "Derivation of global ionospheric Sporadic E critical frequency (foEs) data from the amplitude variations in GPS/GNSS radio occultations" in its current form for publication in Royal Society Open Science.

on behalf of Dr Cécile Lasserre (Associate Editor)
openscience@royalsociety.org

Appendix A

We would like to appreciate the efforts of the reviewers and editor for the comments and suggestions. We have studied all comments carefully and these comments have helped us to improve our manuscript. Following the reviewers' comments, we revised the manuscript. Our responses to the comments and corresponding changes with page and line numbers in the revised manuscript are both detailed below in the blue text. We mark the major changes in the track-change manuscript.

Reviewer: 1

Comments to the Author(s)

This paper describes a method for obtaining sporadic E critical frequencies from satellite GPS radio occultation measurements. This is achieved by using a large network of ground-based ionosondes to correlate with FORMOSAT-3/COSMIC satellite measurements. The authors quickly discover that the correlations (Fig. 1) are not very strong, and do some work to deal with ionosonde measurements close to the instrumental detection limits. An improved correlation is derived, and this will be of value to people using space-based radio occultation measurements to study sporadic E layers.

Response: Thank you for your comments.

The paper contains many grammatical errors and missing references indicated in the text by [?]. In fact, I have rarely been asked to review a paper that has so clearly not been carefully proof-read before submission. Nevertheless, most of it - apart from in a few places - is perfectly understandable.

Response: Thank you for your comments. The references have been updated. The language of the manuscript has been improved with the help of the native-English proof-read.

I have three major concerns:

1. Does any new science come out of this study? All the conclusions about geomagnetic latitudinal dependence and tidally-driven layer descent are already well known (and cited in the text). So are there any discoveries that have come out of this study?

Response: To our knowledge, there is no study of derivation of global Es critical frequencies foEs from the GPS/GNSS radio occultation signals based on a large dataset of ground-based ionosondes and space-based radio occultations. We used a large dataset from COSMIC satellites and 25 ionosondes to derive the hourly foEs. It will benefit the investigation into Es layers.

This study shows that the radio occultation signals can provide reliable Es measurements with rare or no coverage of ground-based ionospheric monitoring stations. The instrumental detection limits of foEs values from ionosondes were found, which should be paid more attention to investigations into weak Es layers by

ionosondes. The geomagnetic latitudinal dependence and tidally-driven layer descent were proven by using the global GPS/GNSS radio occultation signals.

Changes: Please see the abstract.

2. I am not sure that the data really supports your conclusions. For example, the statement on page 7 "The troughs of Es layers with foEs around 2.5 MHz were visible near the magnetic equator and along the 60 – 80 geomagnetic latitude bands" is not borne out by Figure 8. This shows - within the error band - no latitudinal dependence. The same point applies to the statement at the top of page 8.

Response: Thank you for your comments. The statements can be found in Figure 7 (formerly Figure 6) in the global geographical distribution of foEs. The red line in Figure 9 (formerly Figure 8) shows the foEs values with geomagnetic latitudes, in which the Es layer is slightly weaker in the equatorial region. Figure 9 shows the zonal mean foEs values and high northern geomagnetic latitudes of 60°N–70°N. The latitudinal dependence in Figure 9 is less clear than that in Figure 7 particularly in the Southern Hemisphere, likely as a result of the influence of the South Atlantic Anomaly.

3. A second example concerns the apparent tidal descent discussed on page 8 around

line 20. Many of the plots in Figure 9 do not show - from visual inspection - any evidence of descent, either diurnal or semi-diurnal. So you need to explain why at some latitudes/seasons there is evidence of descent, but not in others, or providing a more easily understood and convincing analysis.

Response: Thank you for your comments. The influence of diurnal and semidiurnal tides on the Es layers differs from seasons and latitudes.

The seasonal differences in foEs are due to the dynamics in Es layers, which are directly affected by the wind shear convergence nodes descending with diurnal and semidiurnal tides (Haldoupis, 2006). The diurnal and semidiurnal tides control the descent of Es from 120 km down to 100 km. The influence of the semidiurnal tide is prevalent in June and July, and the influence of the diurnal tide is prevalent in September (Pignalberi, 2014). The transition from the diurnal tide at low latitudes to the semidiurnal tide at high latitudes occurs at 40° latitude (Hays, 1994).

Changes: Please see page 6 lines 40 to 47 in the last paragraph.

Minor points/corrections (this is only a small sub-set - please get a native English speaker to proof read the paper - it should NOT be the job of the reviewers):

Response: The language of the manuscript has been proofread by a native English speaker. Thank you for your comments.

page 4: need to define what the critical frequency and S4max are in the introduction, not later on in the paper.

Response: The description has been moved in the introduction.

Changes: Please see page 3 lines 5 to 12.

page 4: "Whereas, it remains one weak point in explaining the seasonal dependence of Es that is the geographical distribution of Es minimum in the winter hemisphere cannot be simulated by the neutral wind shears.." So, having mentioned this problem, do you solve it in this study?

Response: Not yet in this paper. It is the motivation of this study to derive the global observations from the COSMIC satellites. More extensive observations of Es layers from both ground-based instruments and satellites will help to comprehensively understand the seasonal variation in the Es layer and its mechanism.

Changes: Please see page 2 lines 49 to 50.

page 5: "Besides, the performance of ground-based observations from the world-wide ionosonde network can be evaluated by the comparative results." What does "comparative" mean here - comparing with other ionosondes?

Response: In addition, the quality of ground-based observations from the world-wide ionosonde network can be evaluated by comparing with the global foEs data derived

from FORMOSAT-3/COSMIC RO data. It has been change to express clearly.

Changes: Please see page 3 lines 23 to 25.

page 5: "Six FORMOSAT-3/COSMIC satellites initially followed the same orbit at ~512 km and then sequentially orbit the Earth at ~800 km." This is unclear, what is meant by "sequentially"?

Response: "subsequently".

page 5, line 36: some problem with characters "1500~A ,S2500"

Response: 1500-2500.

page 5: "The ground-truth observations are almost manually scaled or automatic edited values" - unclear

Response: Most ground-truth observations are manually scaled or automatically scaled.

page 6, line 14: "Thses ionosondes"

Response: These ionosondes

page 6, line 47: "the general relationship is loose to be well derived" What does this mean?

Response: Because either detection technique will only measure a subset of the distribution of the intensity of Es layers, the correlation between the two techniques will be influenced by these factors.

Changes: Please see page 4 lines 47 to 49.

page 7, line 37: "The high energy radiation, particle precipitation, and polar gap gravity waves could be responsible for the Es layer at geomagnetic high latitudes." What "high energy radiation" would only affect the atmosphere above 80 degrees geomagnetic? Auroral precipitation occurs in the polar cap, which is at lower geomagnetic latitudes.

Response: Thanks for your comments. Auroral Es layers at high latitudes were long assumed to be a direct manifestation of the aurora. However, in fact, the high-latitude Es layers are much thinner than those produced by auroral precipitation. The windshear drift convergence and the ion vertical transport by the electric-field could be responsible for the Es layers at geomagnetic high latitudes (Bristow, 1991; Kirkwood, 2000).

Changes: Please see page 5 first paragraph of 4. Results.

page 7, line 50: "the standard deviation as the error-in-the-mean" - what else is a standard deviation?

Response: the standard deviation from the mean within a 1° latitude band.

Reviewer: 2

Comments to the Author(s)

Here are some of my comments and suggestion for improvement of the paper.

It has not yet been clearly proven that the lightning has relation to Es layer formation. Some theories and their statistical supports were proposed and prepared. However, the significant issues of those works have been addressed in works of Haldoupis.

Response: Thanks for your comments. This study aims to derive the global foEs of Es layer from the amplitude variations in GPS/GNSS radio occultations. More extensive observations of Es layers from both ground-based instruments and satellites will help to comprehensively understand the variation in the Es layer and its mechanism including the seasonal variations in Es and its relation with tropospheric lightning.

Check misstypings pg.6 ln.15,

Incomplete references pg.5 ln. 40, 41

Missing reference pg.6 ln. 26, 33, 44, 50, pg.8 ln. 12, pg.7. ln. 8, 12, pg.9 ln. 9, 12

Missing reference pg.8, ln. 12, pg.7. ln.8, 12

Response: Thanks for your comments. The references have been updated.

Stations names are difficult to read Fig.1, where the latitude could be found on a particular plot? How the correlation can be proved, what type of correlation/pattern has been found. According to me, there no clear relationship seen. There is no supporting analysis provided for statement:

A general correlation is found between foEs by the ionosondes and FORMOSAT-3/COSMIC S4max at different latitudes.

I would like to point out the fact that problems in scaling foEs close to lower limit of the sounding range is rather related to technical problems due to physical conditions than to the fact that foEs is manually scaled. The physical limits of sounding close to lower ionosonde limit should be discussed instead. Same physical problems affect automatically scaled data too.

Response: Thanks for your comments.

The details of 25 ionosonde stations used in analysis were listed in Table 1. The latitude distribution of stations can be found in the plots of foEs versus geographic latitude or foEs versus geomagnetic latitude and foEs in Figures 8 and 9.

The correlation can be found in scatter plot of foEs and S4max in Figures 2 and 3. We have added Figure 4 of the statistical analyses, which shows the absolute difference and relative difference between foEs from ionosondes and the derived foEs from

COSMIC. The mean and the root mean square error (RMSE) are 0.01MHz and 1.39MHz. A total of 80.61% coincident measurements have a relative difference less than 50%.

The physical limits of sounding close to lower ionosonde limit have been discussed in the revised manuscript. Thanks for your suggestions.

Changes:

Please see page 4 lines 12-13;

Please see page 5 lines 6-14 and Figure 4.

What is the physical meaning of the fitting curves (Fig.2, Fig.3)? How could the use of the linear fitting tell us about the Es layer? And particularly, what would be the use of the interpretation at S4max for values 1.5 or 2? Basically, any fitting can be use any time and some results are obtained. However, there should be some physical background or meaning for type of the chosen fitting.

Response: S4max is an index of an amplitude scintillation resulting from vertical gradients in the ionospheric irregularities. In previous studies, S4max was found to be linearly related to foEs or related to the electron density of ionospheric irregularities (equivalently the square of plasma frequencies f^2) (Whalen, 2009; Arras, 2018). We proposed a method to deal with the outliers of ionosonde measurements close to the instrumental detection limits and gave a better fitting considering the background frequency of the ambient electron density in the absence of Es layers by the equation $(f_oEs - f_{BG})^2 = a \times S4max$. In this way, the ionospheric irregularities within Es layers can be derived from the COSMIC S4max.

Changes:

Please see page 5 lines 1-5.

Interpretation of Fig. 5

foEs values appear to be overestimated by some ionosondes (WU430, SH427 and SA418) as a result of raising a scaling limitation of frequency to avoid the erroneous data close to the threshold of 1.28–1.60 MHz.

Within the text there is no information about multiple layer stratification which may explain the effect as well. It clearly shows the importance of manual scaling where one can easily see the fine structure and distinguish particular layer stratification.

Response: Thanks for your comments. The layer stratification and the importance of manual scaling have been discussed in the revised manuscript.

Changes: Please see page 5 lines 35 to 37.

Correct typos!

The manuscript has been proofread and the typos have been corrected.

References:

- Haldoupis C, Pancheva D. 2006 Terdiurnal tidelike variability in sporadic E layers. *Journal of Geophysical Research: Space Physics* 111.
- Pignalberi A, Pezzopane M, Zuccheretti E et al. 2014 Sporadic E layer at mid-latitudes: average properties and influence of atmospheric tides. 32, 1427–1440.
- Hays PB, Wu D, HRDI Science Team T. 1994 Observations of the diurnal tide from space. *Journal of the atmospheric Sciences* 51, 3077–3093.
- Bristow W, Watkins B. 1991 Numerical simulation of the formation of thin ionization layers at high latitudes. *Geophysical research letters* 18, 404–407.
- Kirkwood S, Nilsson H. 2000 High-latitude sporadic-E and other thin layers—the role of magnetospheric electric fields. *Space Science Reviews* 91, 579–613.
- Whalen J. 2009 The linear dependence of GHz scintillation on electron density observed in the equatorial anomaly. *Ann. Geophys* 27, 1755–1761.
- Arras C, Wickert J. 2018 Estimation of ionospheric sporadic E intensities from GPS radio occultation measurements. *Journal of Atmospheric and Solar-Terrestrial Physics* 171, 60–63.

Appendix B

Reviewer comments to Author:

Reviewer: 1

Comments to the Author(s)

The revised version of the paper is much improved. The points that I raised have been dealt with satisfactorily.

Response: Thank you for your comments.

Reviewer: 2

Comments to the Author(s)

I suggest the authors to return to the all previous comments and read them carefully and after that to make the correction again.

Despite the efforts the authors gave to the work, I do not find it acceptable. It seems that a lot of the suggestions were not answered at all. I suggest the authors to return back to the comments and read them carefully again.

For instance, the limitations of the signal detection by ionosonds has a physical reason, that sounding takes place very close to the gyrofrequency that is given by the geomagnetic field at the specific location. This number varies according to geomagnetic coordinates of the station and is an important part of the particular sounding conditions. The corrections within the text done by the authors are just showing misunderstanding of the physics behind.

I do not find satisfying the answer why and how the interpretations are done.

I do suggest the authors to look at the paper Haldoupis, JASTP, Vol: 172, Pg. 117-121, 2018.

Response: Thank you for your comments.

I would like to explain the point you raised. It should be noted that there is a difference between the limitations of the signal detection by ionosonds as you mentioned and the lower threshold of reliable foEs for manual or automatic scaling in this paper. The former is the lowest frequency of reflection wave recorded in the ionogram (1.0-1.5 MHz) [Haldoupis C., 2011], which is dependent on the sensitivity level of the recording system and absorption in the ionosphere. The sensitivities of different ionosondes differ. The latter is the scaling threshold of reliable values in foEs data identified as the Es layer traces from ionograms, below which it becomes more challenging to distinguish between the Es layer and the background E layer. It can be found that an abnormally high occurrence of Es layers occur within the frequency range 1.28–1.60 MHz. The horizontal red lines in Figure 1 represent the scaling threshold for each ionosonde. A sharp cut-off can be found around the scaling

threshold. It indicates the foEs values are determined less reliably, which don't vary with the S4max values. It is not the lowest frequency of reflection waves due to the physical limits of ionosondes.

Previous studies [Arras and Wickert, 2018; Resende et al., 2018] revealed that S4max is linearly related to foEs or foEs². Based on that, the correlations between S4max and foEs from 25 ionosondes were fitted.

We have carefully read the paper. We have included the reference in the introduction. It is a paper focused on the response of Es layer to thunderstorms. The time delay between thunderstorm activities and the response of Es layers can be explained by the tidal periodicities in the Es variability [Yu et al., 2019].

Change: first paragraph in 3. Data Analysis.

References:

Haldoupis C. 2011 A tutorial review on sporadic E layers. In *Aeronomy of the Earth's Atmosphere and Ionosphere*, pp. 381–394. Springer.

Arras C, Wickert J. 2018 Estimation of ionospheric sporadic E intensities from GPS radio occultation measurements. *Journal of Atmospheric and Solar-Terrestrial Physics* 171, 60–63.

Resende LC, Arras C, Batista IS, Denardini CM, Bertolotto TO, Moro J. 2018 Study of sporadic E layers based on GPS radio occultation measurements and digisonde data over the Brazilian region. In *Annales Geophysicae* vol. 36 pp. 587–593.

Yu B, Xue X, Kuo C, Lu G, Scott CJ, Wu J, Ma J, Dou X, Gao Q, Ning B et al.. 2019 The intensification of metallic layered phenomena above thunderstorms through the modulation of atmospheric tides. *Scientific Reports* 9, 1–13.